

# Research and management challenges following soil and landscape decontamination at the onset of the reopening of the Difficult-To-Return Zone, Fukushima (Japan)

Olivier Evrard [1], Thomas Chalaux Clergue[1], Pierre-Alexis Chaboche[2, 3], Yoshifumi Wakiyama[3], Yves Thiry[4]

[1] Laboratoire des Sciences du Climat et de l'Environnement (LSCE/IPSL), Université Paris-Saclay, UMR 8212 (CEA-CNRS-UVSQ), Gif-sur-Yvette, France

[2] International Research Fellow of Japan Society for the Promotion of Science (Postdoctoral Fellowships for Research in Japan (Standard), Japan

[3] Institute of Environmental Radioactivity, Fukushima University, Japan

[4] French National Radioactive Waste Management Agency (Andra) - Research and Development Division, France

**Correspondence**

Olivier Evrard (olivier.evrard@lsce.ipsl.fr)

**Abstract.** Twelve years after the nuclear accident that occurred at the Fukushima Dai-ichi Nuclear Power Plant in March 2011, radiocesium contamination (with a large dominance of $^{137}$Cs, with a 30-years half-life) remains a major concern in various municipalities of Northeastern Japan. The Japanese authorities completed an unprecedented decontamination programme in residential and cultivated areas affected by the main radioactive plume (8953 km²). They implemented a complex remediation programme scheme relying on different decision rules depending on the waste type, its contamination level and its region of origin, after delineating different zones exposed to contrasted radiation rates. The central objective was not to expose local inhabitants to radioactive doses exceeding 1 mSv yr$^{-1}$ in addition to the natural levels. At the onset of the full reopening of the Difficult-to-Return Zone in Spring 2023, the current review provides an update of a previous synthesis published in 2019 (Evrard et al., 2019). Although this ambitious remediation and reconstruction programme is almost completed, in the 12 municipalities of Fukushima Prefecture in which an evacuation order was imposed in at least one neighbourhood in 2011, from the 147,443 inhabitants who lived there before the accident, only 29.9% of them had returned by 2020. Waste generated by decontamination and tsunami cleaning/demolition work is planned to have been fully transported to (interim) storage facilities by the end of 2023. The cost of the operations conducted between 2011–2020 for the so-called 'nuclear recovery' operations (including decontamination) was estimated by the Audit Board of Japan in 2023 to 6122.3 billion yen (~44 billion euro). Decontamination of cropland was shown to have impacted soil fertility, and potassium fertilization is recommended to limit the transfer of residual radiocesium to new crops. In forests that cover 71% of the surface area of the Fukushima Prefecture and



that were not targeted by remediation, radiocesium is now found in the upper mineral layer of the soil in a
quasi-equilibrium state. Nevertheless, $^{137}$Cs concentrations in forest products (including wood for heating
and construction, wild plants, wildlife game, mushrooms) often keep exceeding the threshold values
authorized in Japan, which prohibits their exploitation in the area affected by the main plume.
Radionuclides from forest were shown to be exported in dissolved and particle-bound forms to downstream
river systems and floodplains, although multiple monitoring records showed the continuous decrease in
radiocesium concentrations in both river water and sediment across the main plume between 2011–2021.
Fish contamination is now generally found below the threshold limits although reputational damage
remains a major concern for local fishing communities. The remobilisation of radiocesium from sediment
accumulated in reservoirs of the region is also of potential concern as it may lead to secondary
contamination of fish or irrigation waters supplied to decontaminated fields. Overall, this synthesis
demonstrates the need to continue monitoring post-accidental radiocesium transfers in these environments
and to keep sharing data in order to refine our predictive understanding of radiocesium mobility and
consolidate the tools available to model contaminant transfers in ecosystems. In forests in particular, novel
countermeasures and wood uses remain to be developed and tested. Furthermore, the hydrologic
connectivity between ecosystems is of great influence on long term radiocesium transport. The
consequences of extreme phenomena (e.g., typhoons, forest fires) that may become more frequent in the
future as a result of global change in these contaminated environments should be further anticipated.

## 1.  Introduction

The Fukushima Dai-ichi Nuclear Power Plant accident (FDNPP) that occurred in March 2011 resulted in the
emissions of large quantities of radionuclides into the environment (Morino et al., 2013). Among these
radionuclides, more than 10 years after the accident, radiocesium is the most problematic substance (mainly for
its longer-lived $^{137}$Cs isotope (T $_{1/2}$ = 30 years) as less than 10% of the shorter-lived $^{134}$Cs isotope (T $_{1/2}$ = 2 years)
emitted initially in similar quantities as $^{137}$Cs was still found in the environment by 2022). The fraction of other
isotopes such as those of plutonium that was supplied by FDNPP accident and that were detected shortly after
March 2011 in environmental samples (2011–2013) were no longer detected during the last years (Diacre et al.,
2023). As the main radioisotope being found in the vicinity of FDNP twelve years after the accident, $^{137}$Cs is a
gamma emitter that is quickly bound to clay minerals (mainly micas such as illite and vermiculite), which show
frayed-edge sites with high selectivity for Cs$^{+}$ cations (Cremers et al., 1988). Possible exposure of organisms to
radiations emitted by radiocesium together with the food chain contamination justified the evacuation and/or the
decontamination of those zones exposed to radiation dose rates exceeding a given threshold (i.e., typically 20
mSv yr$^{-1}$ in emergency conditions and 1 mSv yr$^{-1}$ in 'normal' conditions) (Lyons et al., 2020) (Lyons et al.,
2020) (Lyons et al., 2020) (Lyons et al., 2020) (Lyons et al., 2020) (Lyons et al., 2020) (Lyons et al., 2020)
(Lyons et al., 2020) (Lyons et al., 2020). Although the impact of high radiation doses is debated in Fukushima, a
recent study showed the natural rewilding of the Fukushima landscape following human abandonment, and
suggested that if any effects of radiological exposure in mid- to large-sized mammals in the Fukushima
Exclusion Zone existed, they occurred at individual or molecular scales, and did not appear to manifest (or have
not yet manifested) in population-level responses (Lyons et al., 2020). Another potential issue associated with
$^{137}$Cs is related to the fact that different types of microparticles bearing this radioisotope were found in the





environment and that their inhalation may lead to specific health risks (Hagiwara et al., 2021;Okumura et al., 2019;Miura et al., 2018).

Contrary to the situation observed after the Chernobyl accident where a large (i.e., 30 km-radius zone) contaminated area remained evacuated and abandoned for several decades, resulting in a large scale rewilding of the area (Fesenko et al., 2022), the Japanese authorities decided to conduct ambitious decontamination works in residential and cultivated areas affected by the main radioactive plume (Yasutaka and Naito, 2016). Different zones were delineated to organise decontamination works depending on the initial radioactive contamination

levels found in these zones and the resulting exposition of inhabitants and workers to radiation dose rates. The long-term goal was to keep individual exposure doses below 1 mSv yr$^{-1}$ or 0.23 µSv h$^{-1}$ (reference level of exposure dose in normal times as recommended by the International Commission on Radiological Protection – ICRP). In immediate post-accidental conditions (so-called "emergency exposure situations"), higher dose exposure levels have been allowed (the 20 mSv yr$^{-1}$ level was set in Japan for residence, which corresponds to a

level 3.8 µSv h$^{-1}$ as it takes into account the fact that people spent max. 8 hours day$^{-1}$ outdoors and 16 hours day$^{-1}$ indoors, with indoor dose exposures being 40% of those outdoors). A level of 5 mSv yr$^{-1}$ or 2.5 µSv h$^{-1}$ has also been defined to enforce the individual dose control for workers (e.g. decontamination works, working in forests) in exposed areas of Japan.

As a result of the progressive radioactive decay and the associated decrease in air dose rates with time,

evacuation orders have been gradually lifted in some areas, while they were maintained in the areas exposed to the highest radiation dose rates (i.e., Difficult-to-Return Zone – DTRZ or 帰還困難区域 in Japanese; Figure 1).

The 20-mSv yr$^{-1}$ threshold remained the main guide to delineate these zones, as the so-called DTRZ corresponds to the area where exposure dose was expected to exceed this value even 5 years after the accident (i.e. by 2016). To allow reopening these SDZ and DTRZ zones, decontamination was conducted across wide areas of

Fukushima and neighbouring Prefectures in Japan,  starting with the less contaminated areas (i.e., Intensive Contamination Survey Areas – ICAs; 汚染状況重点調査地域 in Japanese) followed by the Special

Decontamination Zone – SDZ (除染特別地域) (Evrard et al., 2019) and, finally, parts of the DTRZ (Figure 1).



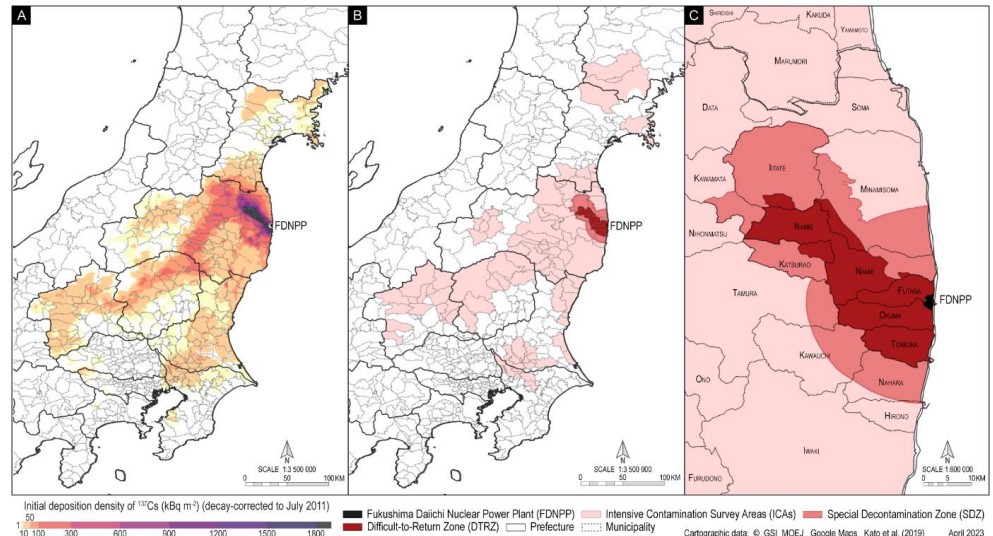

**Figure 1. (A) Map of the ¹³⁷Cs deposition on soils across Northeastern Japan following FDNPP after Kato et al. (2019a); (B) – (C) delineation of the areas where decontamination works have been conducted following FDNPP accident in Japan including Intensive Contamination Survey Areas – ICAs (40 municipalities; 7836 km²), Special Decontamination Zone – SDZ (parts of 11 municipalities; 1117 km²) and DTRZ – Difficult to Return Zone (parts of 8 municipalities; 335 km²) (Source: Japanese Ministry of Environment). The corresponding shapefiles can be freely downloaded from Evrard et al. (2023).**

Decontamination has been completed between 2017–2019 in the ICAs and in the SDZ, with the progressive transfer of remediation waste to interim storage facilities built in Okuma and Futaba Towns (Figure 2). After a partial reopening in 2022, several additional portions of the DTRZ will be reopened from 2023 onwards without obligatory decontamination except in "Special Reconstruction and Revitalization Zones" (特定復興再生拠点区域) (Figure 2). Nevertheless, residential and cultivated zones of the DTRZ located outside of these Specific Reconstruction and Revitalization Zones may be decontaminated as well, in response to the local residents' willingness to see their property remediated.



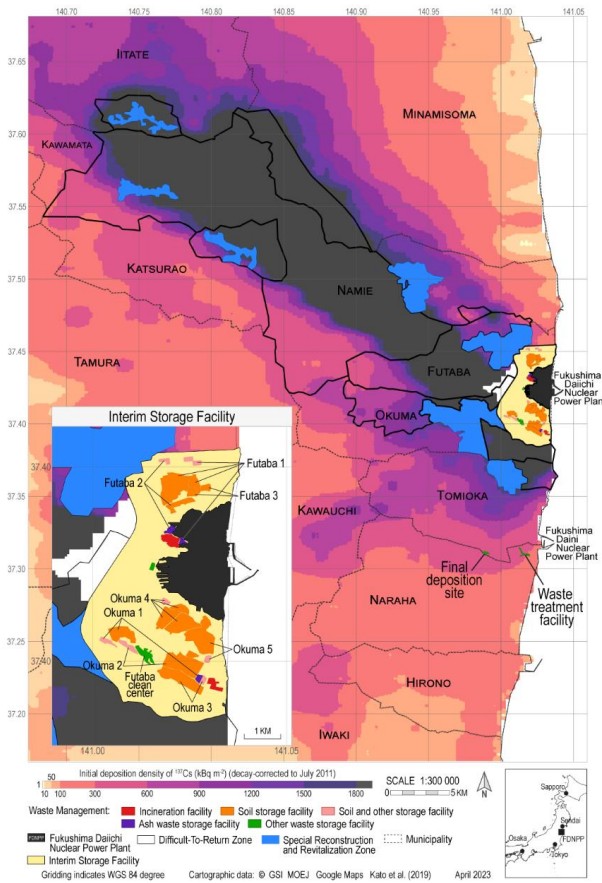

**Figure 2. Detailed map of the reconstructed initial $^{137}$Cs fallout decay-corrected to July 2011 (Kato et al., 2019a), the municipalities of the Difficult-to-Return Zone, the "Specific Reconstruction and Revitalization Zones", and the radioactive waste management and storage facilities (see the inset map for a zoomed view on the location of these facilities). The corresponding shapefiles can be freely downloaded from Evrard et al. (2023).**

According to the definition given by the Japanese Ministry of Environment, a "Specific Reconstruction and Revitalisation Zone" is an area within the DTRZ where environmental rehabilitation projects, such as decontamination and house demolition, are being promoted together with infrastructure development in order to lift the evacuation order. These zones, made possible following Amendments made to the Fukushima Act on Special Measures for Reconstruction and Revitalisation (Act No. 25 of March 31, 2012)(Cabinet Order, 2012), are set up by the Mayor of the municipality, and a plan is prepared and needs approval from the Prime Minister. The plan is being undertaken and completed within five years after approval, and its completion will allow to lift the evacuation order.

In this unique transition context, it is timely to provide an updated synthesis and feedback after an initial review article published in 2019 (Evrard et al., 2019) regarding the completion of this unique remediation programme. The goal is to make the latest information available to the international community and to identify the main



challenges for ongoing and future research at the time when the Fukushima DTRZ will be reopened (i.e. Spring
2023). This review article will therefore not specifically cover the general transfers of radionuclides in different
terrestrial environments, which have been covered by recent comprehensive synthesis articles, nor the fate of
radioactive contamination in coastal and marine waters, which can be found elsewhere (Table 1).

**Table 1. Selection of other review articles and reports dealing with specific radionuclide transfer processes or
remediation techniques.**

_________________________________________________________________________

Topic                                                  Reference(s)

_________________________________________________________________________

Continental transfers of fallout radionuclides          (Evrard et al., 2015;Onda et al., 2020)

Environmental behaviour of fallout radionuclides        (Nanba et al., 2022;IAEA-TECDOC-1927,
2020;Tagami et al., 2022;Nakajima et al., 2019)

Forest transfers of fallout radionuclides               (Hashimoto et al., 2022a)

Impact of radionuclides on freshwater environments      (Nagao, 2021)

Impacts of radionuclides on agriculture                 (Nakanishi and Tanoi, 2016)

Oceanic transfers of fallout radionuclides              (Buesseler et al., 2017)

_________________________________________________________________________

The privileged option is rather to identify the lessons of that unprecedented decontamination programme, as well
as the gaps and needs in remediation actions. After summarizing the calendar for reopening, data on people
having returned and the costs of the remediation works, the challenges for restarting cultivation in the remediated
cropland will be discussed. Then, focus will be laid on decontamination tests and methods in forests, as these
areas covering ca. 75% of the fallout-impact region have not been remediated at this stage (with the exception of
20m-wide buffer strips around houses and roads), and radionuclide cycling remains active in these zones.
Furthermore, the potential export of radionuclides stored in forests will also be addressed, as this may provide a
long-lasting source of contamination to downstream – remediated – or non-decontaminated environments. The
situation in ponds and reservoirs where $^{137}$Cs may be remobilized from sediment will also be examined. Finally,
the main challenges for ongoing and future research will be synthesized.

## 2. Reopening of the zones and return of previous inhabitants

Overall, 165,000 inhabitants were evacuated from the main fallout zone in 2011, and 28,000 people remained
officially considered as evacuees by November 2022 (Reconstruction Agency, 2023). The plans for
rehabilitation of six towns (Futaba, Okuma, Namie, Tomioka) and villages (Iitate, Katsurao) in the DTRZ –
submitted between September 2017 and May 2018 – were approved, and demolition/decontamination are
underway or completed. Evacuation orders have been lifted in Spring 2022 in part of Futaba Town (Futaba
Station area), part of Okuma Town (Ono Station area), Tomioka Town (around Yonomori Station) and Katsurao
Village. Evacuation orders were expected to be lifted in Spring 2023 in similar zones in Namie Town, Tomioka



Town and Iitate Village (Figure 2). According to an extensive report of the Board of Audit of Japan (会計検査

院) published in February 2023 (Board of Audit of Japan, 2023), from the twelve municipalities of Fukushima

Prefecture in which an evacuation order was imposed in at least one neighbourhood in 2011, only 29.9% of the

147,443 inhabitants who lived there before the accident had returned by 2020 (i.e., 44,028 inhabitants). Strong

variations in the proportion of returnees were observed, ranging from less than 5% of the initial population in

municipalities remaining largely closed in 2020 (i.e., Futaba, Okuma, Tomioka, Namie) up to 69.9% in Tamura

(where only a small part was located within the DTRZ).

**3.     Progress and cost of remediation works**

Updated information related to decontamination is regularly provided in Japanese language on a specific website

of the Japanese Ministry of Environment (http://josen.env.go.jp/), although much less information is regularly

communicated in English. For the aggregated costs, information was made available from the above-mentioned

extensive report of the Board of Audit of Japan published in February 2023 (Board of Audit of Japan, 2023).

The management of radioactive waste (i.e., containing at least 8000 Bq kg$^{-1}$ of radionuclides) depends first on

the nature of the material, i.e. whether it consists of soil/vegetation resulting from decontamination works

(referred to as category 1) or whether it consists of debris related to the tsunami or to the demolition operations

in residential settlements (referred to as category 2; Figure 3).

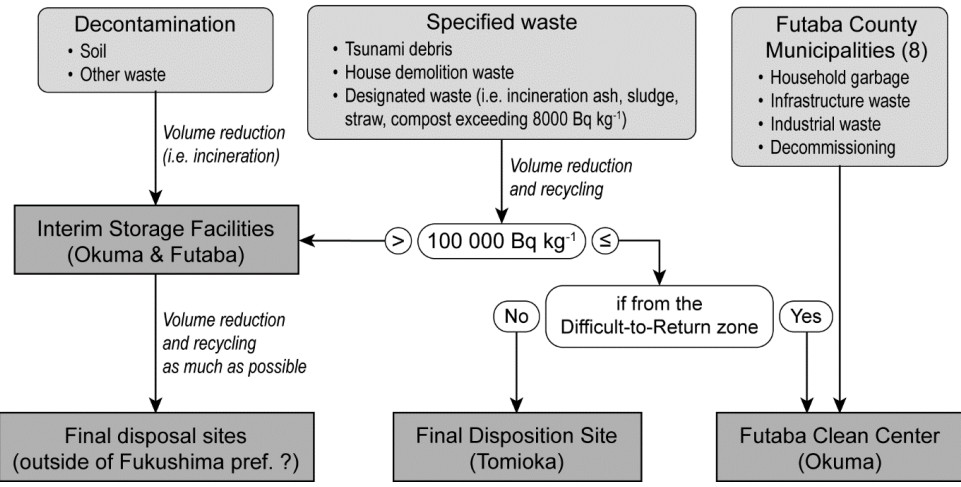

Methods of processing waste containing radioactive materials from Fukushima Prefecture

**Figure 3. Procedures for the treatment of waste containing radioactive materials from Fukushima Prefecture, after**
**information shared by Fukushima Reprun – (Japanese Ministry of the Environment, 2023a)**

For the second category of waste, a threshold of 100,000 Bq kg$^{-1}$ of $^{137}$Cs has been set to determine the disposal

location of this material.

Wastes from the area where countermeasures were implemented (so-called 'specified waste') outside of the

DTRZ and with a $^{137}$Cs content lower than 100,000 Bq kg$^{-1}$ are being stored in a specific disposal site open in





Tomioka Town on 17 November 2017 (Figure 2). For the waste originating from the DTRZ, it is being disposed at the Clean Centre (クリーンセンター) in Futaba Town as decided on 5 August 2022 (Figure 2) (Japanese Ministry of the Environment, 2023b) along with the domestic waste produced in the 8 municipalities of the Futaba County (including Hirono Town, Naraha Town, Tomioka Town, Kawauchi Town, Okuma Town, Futaba

Town, Namie Town and Katsurao Village and covering a surface area of 866 km²).

Regarding the waste resulting from decontamination (soil and vegetation; category 1), as of 7 March 2023 (Japanese Ministry of the Environment, 2023b), the progress of these operations has exceeded 90%, with the latest work being completed in the special reconstruction and revitalization zones of the DTRZ.

Out of a total of 1,372 temporary storage sites scattered across the landscapes in the Fukushima Prefecture, by

February 2023, 31 sites were still storing soil to be removed, 1,341 had seen their waste being completely removed, and 1,070 temporary storage sites had been restored to their original state. 13.43 million m$^3$ of this waste had been transported to interim storage facilities. The soil removed from these sites was sorted and processed, and as late February in 2023, approximately 11.54 million m$^3$ of this waste was stored at the interim storage facilities. Furthermore, 17,382 steel rectangular containers filled with ash dust mainly resulting from

biomass combustion was stored at the same facilities by February 2023. At the end of January 2023, approximately 1.42 million tonnes of waste had been treated at each at the temporary incineration facilities operated by national authorities. Wood crushers are being used to limit the volume of waste from forests (with corresponding volume reduction rates between 45–63%) and combustible waste (i.e. fallen leaves and branches) is being incinerated at these temporary facilities with a very high volume reduction (96–99%) and a very low

transfer of $^{137}$Cs to exhaust air (≤0.3 Bq m$^{-3}$) (Hashimoto et al., 2022b). Special attention is nevertheless being required for the subsequent management of combustion ash with high $^{137}$Cs concentrations (> 2,000,000 Bq kg$^{-1}$).

In accordance with the Act on Special Measures Concerning the Handling of Contamination by Radioactive Substances, the specified waste containing more than 100,000 Bq kg$^{-1}$ of $^{137}$Cs is also being stored at these

interim storage facilities in Okuma and Futaba towns that became operational in October 2017. These facilities covered a total area of 1285 ha acquired from 1853 land owners by late February in 2023 (orange surface areas on Figure 2). According to the Japanese law, this material is supposed to be transported to final disposition sites located outside of the Fukushima Prefecture at locations that remain to be selected by 2047 (i.e., 30 years after the interim storage facilities became operational).

The costs of the recovery and reconstruction operations conducted by the Japanese authorities following the Great East Japan Earthquake and the associated nuclear accident at FDNPP during the period comprised between 2011 and 2020 were recently synthesized by the Board of Audit of Japan in February 2023 (Board of Audit of Japan, 2023). A conversion rate of 1 EUR = 140 yen (as of January 1$^{st}$, 2023) was used in the current text. They estimated that the total of these costs reached 38,171.1 billion yen (~273 billion euro), with a provisioned and

remaining budget of 6144 billion yen (~44 billion euro). Among these costs, 20% of the budget (7745.6 billion yen; ~55 billion euro) was devoted to reconstruction/public works and 16% of the budget (6122.3 billion yen; ~44 billion euro) was spent for the 'nuclear recovery policy' including decontamination. Most of the provisioned and unused budget by 2020 is expected to be spent on reconstruction/publics works (34% of residual budget;



2094 billion yen; ~14 billion euro) and nuclear recovery (22%; 1343.9 billion yen; ~9.6 billion euro). For the
period between 2011–2020, as the annual budget of the State of Japan varied from ~90,000 to ~100,000 billion
yen /year (~700 to 800 billion euro), this demonstrates that the budget spent on the reconstruction following the
Great East Japan Earthquake and the associated nuclear accident at FDNPP was far from negligible.

### 4. Recultivation of cropland

In the least contaminated cropland fields (with typical $^{137}$Cs activities < 5000 Bq kg$^{-1}$), tillage and topsoil/subsoil
interchange provided a common countermeasure (Evrard et al., 2019). A study conducted at an experimental site
of the ICA zone (Ibaraki Prefecture) between 2011–2017 confirmed that $^{137}$Cs concentrations in both soils and
crops (i.e., soybean) decreased exponentially with time elapsed since the accident as a result of tillage operations
(Li et al., 2019). In contrast, in paddy fields and other cropland with $^{137}$Cs levels exceeding 5000 Bq kg$^{-1}$
(covering a surface area of ~827 km² in Fukushima Prefecture)(Nakanishi, 2018), a major countermeasure for
soil decontamination has generally consisted in the removal of the 5-cm upper layer concentrating radiocesium.
In the SDZ and DTRZ, the soil removal was compensated by the addition of crushed granite and/or saprolite
extracted locally and its mixing with the residual initial soil profile (Evrard et al., 2019). In the official reports
and guideline books, this practice is justified by the need to supply "fresh", "clean" or "new" soil "in order to
ensure the conditions that enable resumption of agricultural production are restored"(Japanese Ministry of
Environment, 2013). However, this decontamination process may have led to a decrease in fertility of these soils
and enhanced erosion, with strong heterogeneities within the fields (Inoue et al., 2020). Technical developments
are currently under progress to contribute to the rapid assessment of soil fertility in this context based on
hyperspectral reflectance measurements (400–2500 nm) of soil samples, and the subsequent calculation of
spectral index algorithms (Inoue et al., 2020).


Another commonly applied countermeasure is the application of potassium (K) fertilizers in order to promote the
K-Cs (as an analogue) competition at the soil solution-root interface and reduce the root uptake of radiocesium
by crops (Zhu and Smolders, 2000). The positive effect of potassium fertilizer on radiocesium transfer to
vegetation has been widely studied in post-Chernobyl studies; that countermeasure is most effective in soils with
naturally low levels of exchangeable potassium. As a remedial action in contaminated soils of Fukushima
Prefecture, increasing exchangeable potassium up to 25 mg K$_2$O 100 g$^{-1}$ in soil was recommended for
agricultural soils where the concentration of exchangeable potassium was inferior to that level (Japanese
Ministry of Agriculture, 2013). Furthermore, recent research based on field experiments demonstrated that a
complementary soil property (i.e. the non-exchangeable K) and a level < 50 mg K$_2$O 100 g$^{-1}$ could be used as
another threshold for use along with that of exchangeable K (<25 mg K$_2$O 100 g$^{-1}$) to identify soils that would
need additional K fertilization (Kurokawa et al., 2020). Another field of research is devoted to the optimization
of agricultural practices for the K budget. For example, Nishikiori et al. (2020) investigated the plant-available K
budget at the field scale, by comparing two fields with different soil textures and drainage conditions. The major
inputs of K to the fields were shown to be fertilization, straw return and irrigation, while the major outputs were
plant harvesting, surface runoff and water percolation. Nevertheless, most K harvested with the plants (85%) was
brought back to the soil by straw return. In contrast, water percolation and surface runoff were the dominant
output pathways, with most of K being discharged from the fields before mid-summer drainage, contributing
significantly to the general negative K balance comprised from -20 to -289 kg ha$^{-1}$). A careful irrigation adapted



to the soil conditions is thus recommended to reach a more appropriate K budget in soils in order to limit the
radiocesium transfer to the plants.

The potential transfers of radiocesium via irrigation water in paddy fields also remain a matter of concern. The
monitoring of total and dissolved radiocesium in irrigation water, rice and soil from two decontaminated paddy
fields showed that 85% of radiocesium in irrigation water was not exported and remained in the field (Shin et al.,
2019). However, the quantity of additional radiocesium supplied by irrigation water was negligible (~0.08% of
the initial inventory) compared to the initial supply by radioactive fallout in 2011. This resulted in very low soil
to brown rice transfer factors of radiocesium (0.0015–0.0068).

## 5. Radionuclide cycling in forests and contamination of forest products

The situation in forests should receive a particular attention, as they cover 71% of the surface of Fukushima
Prefecture (ca. 970,000 ha) (Hashimoto et al., 2022a). Evergreen coniferous trees (Japanese cedar – *Cryptomeria
japonica*, Japanese cypress – *Chamaecyparis obtusa*, red pine – *Pinus densiflora*) used for construction timber
account for about 40% and deciduous trees (konara oak – *Quercus serrata*) – mainly for papermaking materials
and mushroom cultivation – for about 60% (Hashimoto et al., 2022d). Before FDNPP accident, the Abukuma
Highlands were actively used for cultivating trees (mainly konara oak) for producing mushroom logs.

The FDNPP accident atmospheric fallout was at the origin of the contamination of 2600 km² of forests with
radiocesium levels exceeding 100 kBq m$^{-2}$, which should be compared to the 360-km² surface area of cropland
affected by those levels (Onda et al., 2020). Overall, 60–90% of radiocesium that fell on cedar and cypress
forests was first intercepted by the canopy (leaves and branches) and partially absorbed by the foliage. In
contrast, the leaves of deciduous trees had not burst yet, and the level of fallout trapping was therefore lower in
broadleaf forests (Hashimoto et al., 2022d;Kato et al., 2019b). Then, radiocesium was rapidly transferred by
water (throughfall and stemflow) and defoliation (litterfall) towards the forest floor and the underlying mineral
soil, which represented the most important pools of radiocesium a few years after the accident. Vertical
redistribution of radiocesium between forest floor and mineral layers was established within 1-2 years after the
fallout and changed slowly thereafter. The rapid accumulation in topsoil layers is likely related to the relatively
wet climate, which promoted the early vertical migration of radiocesium in numerous Japanese forests. In the
mineral soil layer, radiocesium is firmly retained on mineral soil particles and its content peaks at shallow depths
(typically < 5 cm) (Hashimoto et al., 2022d). However, despite the current low migration with depth,
bioturbation by organisms may also contribute to further soil and associated radiocesium transfer across the soil
profile.

In contaminated forests, both the forest floor and the topsoil layers now represent significant reservoirs of
radiocesium available for root uptake by trees and understory species. That phenomenon is the predominant
cause of the long-term radiocesium recycling associated with the biomass turnover and that of a possible long-
lasting contamination of forest products (Goor and Thiry, 2004). Variability in stemwood contamination (i.e., the
trunk excluding the bark) is expected to depend on various factors: time after fallout, tree species and age,
position in the stand, radiocesium and K contents in the soil, etc. (Ohashi et al., 2020). Initial vertical and radial
movement of radiocesium within the stemwood and its redistribution between sapwood and heartwood has been
variable depending mainly on tree species (Ota and Koarashi, 2022). Radiocesium content and distribution in



stemwood of both coniferous and deciduous trees is now approaching the equilibrium state, although a higher
$^{137}$Cs concentrations in heartwood of cedars compared to oaks remains under investigation (Hashimoto et al.,
2022d). An aggregated transfer factor for different major tree species has been proposed to calculate the tree

organs contamination in radiocesium taking into account the total surface contamination levels estimated by
airborne surveys (Hashimoto et al., 2020c). However, the proportion of radiocesium remaining in the forest floor
of different forests remains quite variable (Imamura et al., 2020), which may induce uncertainty in further
recycling by trees as radiocesium found in the organic layer (devoid of clay minerals) is usually much more
bioavailable for root uptake (Thiry et al., 2000). A field monitoring of radiocesium bioavailability in soils was

conducted at two neighboured forest sites (one with Japanese cedar, the other with konara oak) of Kawauchi
Village between 2011–2017 (Manaka et al., 2019). An exponential decrease in the proportion of exchangeable
$^{137}$Cs was observed in organic and mineral soil layer samples at both sites. The proportion significantly
decreased within 2–4 years after the accident, becoming almost constant thereafter (2–4%). These results support
the interpretation that contaminated forests have entered a steady-state phase of $^{137}$Cs cycling. Several articles on

tree contamination with time have been published in recent years and they all confirm similar findings, which
increases our confidence in these results (Gonze et al., 2021;Yoschenko et al., 2022).

The fate of radiocesium cycling in forests can also be anticipated with numerical modelling. Recent simulations
showed that initial foliar absorption by coniferous trees and subsequent internal transfers promotes the
radiocesium persistence in trees and thus have a strong possible impact on the early phasing of tree

contamination (Thiry et al., 2020;Thiry et al., 2018). The simulated contribution of foliage and root uptake to the
tree contamination were equivalent 10–15 years after the atmospheric deposits, but the further root uptake was
too low to compensate the activity decline in the tree with time. In a model inter-comparison for Japanese forests
(Hashimoto et al. (2020a), convergent simulations confirmed that an equilibrium state in tree wood
contamination is reached ca. 10 years after the initial fallout, even in konara oaks, and that they then decline

slowly. The highest uncertainty in the simulations of wood contamination remains for newly planted trees,
where root uptake is the only pathway. These results indicated that the parametrization of long term net root
uptake remains uncertain without additional field monitoring.

Wood contamination in particular represents a problem for log production for mushroom cultivation and fuel
chips production, not only in Fukushima Prefecture but also in neighbouring Prefectures of Japan (due to the

lower $^{137}$Cs limits – 40 or 50 Bq kg$^{-1}$ max. – allowed for these uses of wood) (see Table 2). Debarking of trunks
can lower $^{137}$Cs concentrations to allow the use of the wood for biofuel production, although care will have to be
taken to manage the resulting ashes (Hashimoto et al., 2022b). A technology for methane fermentation of
contaminated wood biomass has also been tested (Hashimoto et al., 2022b), with the methane gas produced
devoid of radiocesium, which was recovered instead in the fermentation digestate. However, decontaminating

digestate before its use as a common fertilizer remains problematic (Kobayashi et al., 2020). New challenges
may involve the management of abandoned contaminated fields and certain waste substrates using new forests or
short rotation coppice dedicated to bio-fuel production. That implies the *in situ* radiological control of
radiocesium cycling on long timescales through appropriate and sustainable methods of biomass cultivation
adapted to the Japanese ecological conditions and its conversion into energy as it was tested in post-Chernobyl

studies (Vandenhove et al., 2001;Thiry et al., 2001) and, more recently, in post-Fukushima experiments



(Kobayashi et al., 2013). Tree logs soaking and/or further wood chips/sawdust washing in presence of a suitable radiocesium absorbent (e.g. Neda (2013) represent other potential countermeasures, which still need to be tested or improved for the Fukushima conditions. For Chernobyl contaminated forests, the study of Shaw et al. (2001) revealed that a cost-effective management strategy would also require novel alternative uses of forest products,

which could provide added value to the standing crop in return for small increase in public and worker doses. In that context, Dubourg (1996) recommended a radiological clean-up approach involving both the incineration of the most contaminated parts of the tree and the branches, and the transformation into paper pulp of the less contaminated part of the trunk. Another idea that emerged after the Chernobyl accident but that was never tested (personal communication: Y. Thiry) would imply a shift in the local use of wood products through e.g. the

production of wood shingles after a special treatment of standing coniferous trees through debarking. As practiced in Scandinavia from the early medieval period, the debarking provokes the decline of the tree together with the resination of the stemwood. After several years of tree decline, the natural impregnation of the timber is supposed to be associated with a loss of potassium in stemwood and by analogy with a removal of radiocesium. Shingling (*itabuki* - 瓦礫) using cedar wood was widely used in forest areas of Japan and even in urban areas

until the end of the Edo period (1603–1867)(Japanese Architecture and Art Net Users Systems, 2001). Hopefully, by promoting a natural decontamination of wood, such a practice, once adapted, could also provide a source of added-value and revitalization for the local wood industry in contaminated forest areas.

**Table 2. Current radiocesium threshold values for forest products in Japan, after Hashimoto et al. (2022d).**

_______________________________________________________

| Forest product | Maximum radiocesium concentration (Bq kg$^{-1}$) |
| --- | --- |
| Tree logs for shiitake cultivation | 50 |
| Sawdust for mushroom cultivation | 200 |
| Firewood (for cooking) | 40 |
| Charcoal (for cooking) | 280 |
| Wood pellets | 40 |
| Bark compost for livestock bedding | 400 |

_______________________________________________________

Mushroom and wild plant contamination is still particularly problematic in a country where the hunt for

mushrooms and sansei (i.e. wild/mountain vegetables) is so popular both as a popular leisure activity or part of cultural traditions (offerings for Bon Festival in August or New Year's Day). Satoyama (里山) traditions are widespread in mountainous rural zones located in the vicinity of forests in Japan where people live in harmony with nature and collect edible food from forests (including plants – referred to as sansai or 山菜 – and mushrooms). Because of the high contamination found in forests and its static character, the absence of

decontamination and the ability of these organisms to absorb radiocesium efficiently, they represent the vast majority of food products exceeding the standard limit of 100 Bq kg$^{-1}$ (Hori et al., 2018), with the dominance of



wild plants in Spring and that of mushrooms in Autumn (Hashimoto et al., 2022b). As of Nov. 2020, shipping restrictions on mushrooms had been imposed by 117 municipalities in 11 Prefectures (Figure 4).

It is impossible to provide a comprehensive list of species to avoid collecting, as there are 4000 – 5000 species of
wild mushrooms in Japan. Nevertheless, a study showed that although 76% of the mushrooms collected between 2016-2019 in Kawauchi Village located nearby the DTRZ exceeded the maximum allowed threshold of 100 Bq kg$^{-1}$ in $^{137}$Cs, the committed effective dose due to consuming mushrooms was lower than 1 mSv per year (Cui et al., 2020).

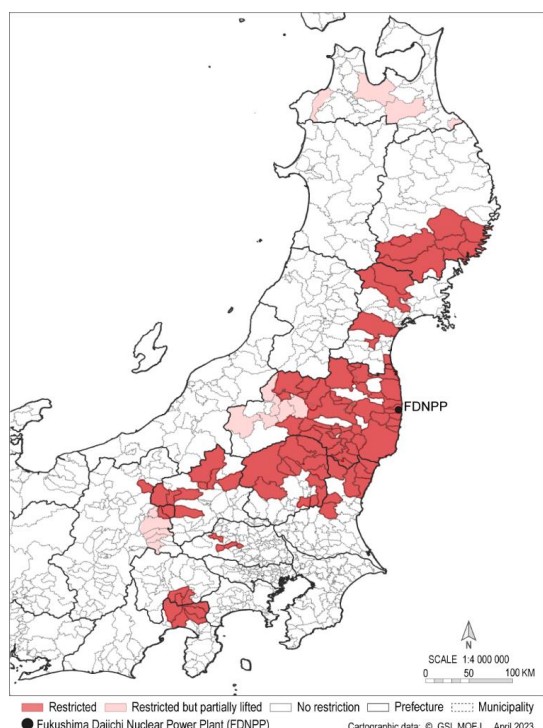

**Figure 4. Map of the Japanese municipalities with mushroom shipping restrictions as of November 2022 after Hashimoto et al. (2022b). The corresponding shapefiles can be freely downloaded from Evrard et al. (2023).**

$^{137}$Cs concentrations in mushrooms were found to be related to the soil contamination, and normalized concentrations allow comparisons between regions/species. In general, as observed in Chernobyl-contaminated forests, mycorrhizal fungi living in symbiosis with trees have higher $^{137}$Cs concentrations than saprotrophic
fungi, obtaining their nutrients from decomposing dead wood and leaves (Komatsu et al., 2019). However, this general result was counterbalanced by the fact that some species showed a different behaviour, which justifies the general prohibition of all mushrooms in a wide area across Northeastern Japan as a matter of precaution. In a similar way as for wild mushrooms, $^{137}$Cs concentrations were found to be strongly variable in wild plants, with some plants showing noticeably higher contamination levels, e.g. koshiabura (*Eleutherococcus sciadophylloides*)
and some cooking techniques were found to be effective to decrease $^{137}$Cs levels in wild plants, e.g. Hashimoto



et al. (2022b). There has also been a large impact in recreational activities, with a large decrease in the number of climbers and fishers visiting the Fukushima Prefecture, along with a similar decrease of urban visitors.

In addition to wild mushroom picking, mushroom cultivation is very popular in Japan and it represented 80% of the non-wood forestry production in 2017 (i.e. the non-wood forestry production excluding timber, which

accounted for 43% of the total forestry production in Japan). Two techniques are being used, i.e. bed-log cultivation (mainly used for *shiitake* production) or sawdust medium cultivation, with the second technique becoming increasingly dominant in recent decades in Japan (as it is less man-power consuming and given the increasing preference of the Japanese population for mushrooms other than *shiitake*). This trend was further accentuated after the FDNPP accident, with the restriction of the production and shipment of shiitake

mushrooms across a wide area in Japan. The problem is that wood (mainly of konara oak – *Quercus serrata*) from the Fukushima Prefecture (e.g. Abukuma Mountains) was not only used for the local production of *shiitake* but also to provide the mushroom logs used for the production carried out in other Japanese Prefectures. This production had to be stopped because of the very high levels of $^{137}$Cs (up to more than 10 times the index value) recorded in the production region (Hashimoto et al., 2022b). Interestingly, as observed for cultivated soils, a

strong negative correlation was observed between radiocesium concentrations in the most recent branches of konara oak and exchangeable K in the soil surface (0–5 cm depth) layer (Kobayashi et al., 2019). Since mushroom log production requires a 20-year cycle, a long term monitoring may open the way to a better use of konara oaks grown on soils with high concentrations of exchangeable K to produce less or even uncontaminated wood.

**6.  Decontamination of forests**

Forest decontamination methods can be subdivided into two groups: (i) the measures actually implemented in Fukushima-impact zones and (ii) those evaluated for research or practical issues. As of (i), only the area located within 20 m of the forest edges (bordering residential areas, roads and other living areas) were treated. In this buffer zone, the vegetation and contaminated organic layer (litter and humus) is gathered and transported out of

forests. The removal of the forest floor litter layer had soon been considered as a potential method for reduction of tree contamination in experimental studies. Removal of forest floor was likely to be more useful when operated after the peak of transfer of radiocesium from the above ground parts of trees to the organic layer (Thiry et al., 2018). That peak typically occurs around 3–5 years after the initial deposition, although the time period varies with tree species and soil organic layer characteristics (IAEA-TECDOC-1927, 2020). Koarashi et al.

(2020) tested the impact of litter removal conducted in a broadleaf forest in July 2014 (i.e., more than 3 years after the accident). They observed no effect on tree contamination and a decrease of litter $^{137}$Cs contamination in the 1$^{st}$ year following remediation and suggested that this method should be applied even more rapidly (within 1–2 years) after the accident, before the significant transfer of $^{137}$Cs from litter to the underlying mineral soil. For coniferous stands, Thiry et al. (2018) indicated a low response of tree contamination to litter removal even in the

long term, because the initial $^{137}$Cs foliar absorption is more influential than the root pathway for a long time.

In addition, clear-cutting (i.e. cutting all trees) and thinning (i.e. partial cutting) can be implemented, however, their additional contribution to decrease air dose rates could not be demonstrated. Potassium fertilization can therefore be used to limit the absorption of radiocesium by the plants, as demonstrated for konara oaks (Kobayashi et al., 2019).



In parallel to these decontamination issues, it should be stressed that the absence of forest management in man-made forests can lead to other problems, such as the spread of insects and diseases. Forest maintenance includes planting, clearing, thinning and maintenance of forest roads, which are crucial to avoid a degradation of different forest functions (e.g. carbon sequestration, wood production, landslide control). Moreover, thinning and other particular silviculture treatments remain important for reducing fuel load and fire hazard, or just for

implementing strategic fuel breaks, as experimented in Chernobyl forests (Ager et al., 2019). Still, the forest area which has been maintained in the Fukushima Prefecture has decreased by ca. 50% from 2011 onwards as a result of the access restrictions due to high dose rates (Hashimoto et al., 2022b). Restrictions have also been imposed on the shipping of wildlife meat (i.e. wild board, Asian black bear, Sika deer, Spot-billed duck, green pheasant, copper pheasant) in the Fukushima Prefecture and/or in nearby Prefectures because of their excessive muscle

contamination in [137]Cs. These shipping restrictions along with a decrease of the hunter numbers observed all across Japan led to wildlife population expansion. Wild boar proliferation, in particular, leads to extensive damage to houses and crops in the main [137]Cs-fallout-impacted area (Hashimoto et al., 2022b). A recent study analysed [137]Cs concentrations in wild boar muscle samples (n=221) collected from the DTRZ and surrounding areas between 2016 and 2020. This research outlined higher activity concentrations observed in the DTRZ

compared to the surrounding areas, and an overall decrease of [137]Cs values with time (Saito et al., 2022). Seasonal variations in [137]Cs muscle concentrations were also observed, and these may be related to changing food habits and the fractions of available [137]Cs in the material ingested by wild boars (Saito et al., 2020). Nevertheless, these seasonal variations were shown to be less pronounced than in wild board contaminated in Germany following the Chernobyl accident, and this observation may be due to their more diverse food sources

for wild boards in Japan (Berendes and Steinhauser, 2022). Overall, as the [137]Cs contamination in wildlife muscles decreases with the distance from FDNPP, the strategy may be two-fold. First, in the areas farther from FDNPP where muscle contamination is likely to remain below the standard limit of 100 Bq kg[-1] for meat consumption, shipping of the meat may be authorized after inspections of all slaughtered individuals conducted at special facilities specifically designed to this end in Tochigi and Ibaraki Prefectures. Second, in the areas close

to FDNPP where this threshold is expected to be exceeded for a long time and where wildlife population expands, active extermination (by hunting or capturation) and the subsequent incineration of the bodies should be considered (Hashimoto et al., 2022b).

## 7. Export of radionuclides from forest to riverine ecosystems

Forests that remain contaminated with [137]Cs may supply contamination to lower landscape areas. A study
showed that dissolved [137]Cs concentrations measured in a stream draining a forested headwater catchment was mainly derived from soil water with high dissolved [137]Cs concentrations originating from litter leachate. When storms occur, with the expansion of soil saturated zones, an increase in dissolved [137]Cs concentrations coinciding with the release of water stored in shallow soil layers is observed (Iwagami et al., 2019). This additional supply of dissolved [137]Cs from forest litter has been confirmed by leaching tests conducted on broadleaf litter in an area

affected by saturation overland flow during storm events (Sakakibara et al., 2021).

Different pathways of radiocesium transfer from forests to river systems can be found, i.e. via litter fall into rivers, lateral inflow from the forest litter layer, and lateral transfer from the underlying forest soil. In a modelling exercise, Kurikami et al. (2019) showed that the decreasing trend of [137]Cs in river water and





freshwater fish was due to a combination of the decreasing contamination trend in the forest leaves/needles and litter compartments, and the increasing contamination trend in soil.

When clearcutting is conducted, suspended sediment exports were found to increase two-fold, with a much more limited increase in [137]Cs export due to the very high sediment contribution of areas with low [137]Cs concentrations, e.g. channel bank erosion (Nishikiori et al., 2019).

Another approach relied on the use of a mass balance model to map the spatial distribution of [137]Cs inventories and quantify [137]Cs transport via sediment and litter in 2016–2017 along a deciduous forested hillslope of Date, in the Fukushima Prefecture (Oda et al., 2022). They showed that [137]Cs inventories were significantly higher in downslope riparian areas (455 kBq m[-2]) than in the upslope ridge area (179 kBq m[-2]). Annual [137]Cs transport with litter and sediment corresponded to less than 0.5% of the [137]Cs hillslope inventory, and transport of litter with high [137]Cs activity concentrations was found to provide the main pathway of [137]Cs transfer at that scale.

Nevertheless, these transfers of [137]Cs in forest environments, although significant in terms of export of contaminated material, were not shown to have an impact on radiation dose rates in forests, as shown by the monitoring of radiation levels along a hiking trail across forests in Tomioka Town in 2019, which remained stable despite the occurrence of Super Typhoon Hagibis in October 2019 (Taira et al., 2020).

## 8. Impacts of decontamination in radionuclide activities in riverine systems

Multiple recent publications confirmed that [137]Cs concentrations in river water and in sediment transported in rivers draining the main radioactive pollution plume strongly decreased between 2011 and 2020, and some of these datasets are available in open access (Taniguchi et al., 2020;Evrard et al., 2021). A database compiling [137]Cs activities measured in sediment (n=782) collected from 27 to 71 locations during 16 fieldwork campaigns conducted between November 2011 and November 2020 across catchments (6450 km²) draining the main radioactive pollution plume of the Fukushima Prefecture demonstrated that the radiocesium levels in sediment transiting these rivers decreased by more than 90% between 2011–2020 (Evrard et al., 2021).

Interestingly, very similar results were obtained based on continuous and more detailed monitoring in local upper catchments of the region. Fluvial discharge of [137]Cs was monitored between 2011 and 2021 from two small rivers including the Hiso River draining mainly farmland (4 km²), and the Wariki River, draining mainly forests (7 km²), in Iitate Village (Ueda et al., 2021). Both particulate and dissolved [137]Cs concentrations – particulate fluxes representing 90% of the total [137]Cs export – were shown to have decreased very strongly by more than 90% over a 10-year period (with higher decreases observed in the catchment dominated by farmland than in that dominated by forests).

In the main river of the region (i.e. Abukuma River, draining ca. 5300 km² of land characterised by heterogeneous initial [137]Cs deposition levels), Taniguchi et al. (2019) found that the high [137]Cs concentrations observed in suspended sediment just after the accident in 2011 showed a steep exponential decline that lasted for about one year and that was dominated by the supply of [137]Cs from paddy fields, other farmland and urban areas. This initial phase was followed by a more gradual secondary decline, with a higher contribution of [137]Cs from forests. Overall, the particulate form of [137]Cs represented 96.5% of the exports investigated in this study between June 2011 and August 2015.



Decontamination works conducted in farmland and residential areas took place from 2013 to 2018 in the Special Decontamination Areas, including in those areas located nearby the Niida River flowing across Iitate Village and Minamisoma Town. A study combining river monitoring with governmental decontamination data and high-resolution satellite images provided a comprehensive impact assessment of these remediation works. Feng et al.
(2022) showed the occurrence of two phases, with a first stage of increase of erosion (+237%) during decontamination (2013–2016) – when soils were left bare to remove the topsoil surface layer concentrating [137]Cs – followed by a decrease during the subsequent revegetation stage (2016). Despite this higher sediment supply, they showed that the material delivered to river systems contained reduced [137]Cs levels compared to the pre-decontamination period and that this stage of higher sediment supply and transfer was only temporary due to the
rapid vegetation recovery after the completion of remediation works.

Typhoon Hagibis that made landfall in Japan on October 12, 2019 (Irasawa et al., 2020) was the first extreme rainfall event that occurred in the region after the completion of decontamination in the Special Decontamination Areas early in 2019. Its impact on sediment sources and sediment [137]Cs contamination was investigated through the analysis of flood sediment deposits collected in the Mano and Niida River catchments and to the comparison
of their geochemical and colour properties with those analysed in potential sources (e.g., cropland, forests, and subsurface material originating from landslides and channel bank erosion) (Evrard et al., 2020). The results showed that cropland and forests provided the main supply of sediment, and that these sources had reduced [137]Cs concentrations, which may be explained by the effective decontamination of cropland and the dominance of rill and gully erosion under forests – mobilising deeper soil layers depleted in [137]Cs – after such an intense event.

A potential concern was that sediment from forests – with high [137]Cs concentrations as these zones have not been remediated – that may be supplied to river systems during heavy rainfall events and deposited in nearby cropland – that has been remediated – due to river overflow during intense hydrological events – may lead to a recontamination of these areas. To investigate this potential issue, sediment that had deposited after the flood generated by the 2019 Hagibis typhoon was collected along two rivers in Iitate Village to determine the total and
exchangeable [137]Cs, and acid-extractable potassium (K) contents (Asano et al., 2022). These parameters were compared to those measured in nearby decontaminated soils (where no flood sediment deposition had occurred). Although sediment deposited by the flood showed 4 times higher [137]Cs concentrations than decontaminated soils, it showed a 3-times lower [137]Cs exchangeable content. Furthermore, acid-extractable K referred to as non-exchangeable K was found to be sufficiently high to restrict [137]Cs transfer from soil to crops that may be planted
to these fields afterwards.

Another issue may be related to the intermittent storage of [137]Cs contaminated sediment in the river floodplain that may be remobilized in the future during extreme flooding events, although their residence times remain currently uncertain (Golosov et al., 2022). This will depend on the floodplain morphology, vegetation characteristics and on the planning of management operations (as river channels may be dredged and cleaned by
the authorities). Overall, another study (2015–2019) conducted to determine changes in radioactive air dose rates in two riverside parks (along the Mizunashi River) of Minamisoma City showed a general decrease in these rates. They attributed 35% of the reduction to the physical decay of radiocesium, 14% to the vertical migration of radiocesium into the soil, and 51% to the combined effect of typhoons and remediation works between 2015–2019. They outlined in addition the great attenuation of air dose rates due to the strongest typhoon Hagibis in



2019, which generated a significant flush of contaminated sediment stored in the plain towards the Pacific Ocean (Yamasaki et al., 2023).

As fishing was a major recreational activity in the region (which is allowed after purchasing a fishing ticket from a cooperative), the emergency monitoring of wild and cultured freshwater products by the Fukushima Prefecture was implemented as soon as on March 30, 2011, with the exception of the designated evacuation zone (now referred to as the Difficult-to-Return Zone), which had not been targeted by monitoring inspections because of the expected high $^{137}$Cs activities in freshwater products (Wada et al., 2022). For cultured fish in ponds (mainly common carp, salmon and char), very few samples exceeded the Japanese regulatory limit of 100 Bq kg$^{-1}$ of $^{137}$Cs in 2011–2012, and these concentrations were found to be below the detection limits of $^{137}$Cs (~7 Bq kg$^{-1}$) in all samples from 2015 onwards. This may be explained by the fact that radiocesium uptake from food was controlled (using non-contaminated pellets and setting an intake screen from preventing contaminated wild prey to enter from outside into the ponds). In contrast, several freshwater fishes were contaminated (containing more than 500 Bq kg$^{-1}$ of radiocesium and up to 18,700 Bq kg$^{-1}$ in 2011–2012), and the 100-Bq kg$^{-1}$ regulatory limit was no longer exceeded in freshwater fish by 2020 onwards (with the exception of the Difficult-to-Return Zone as it was not covered by the monitoring inspections). This justified the prohibition of shipment of eight species (ayu, common carp, crucian carp, Japanese dace, masu salmon, white-spotted charr, Japanese eel, Japanese mitten crab) from some areas of the Fukushima and four neighbouring Prefectures as of April 2021 and the ongoing prohibition of fishing in the Difficult-to-Return Zone (where salmon and char fishes showing contamination levels of up to 25,006 Bq kg$^{-1}$ were analysed in 2016)(Wada et al., 2019). Although fishing has been allowed again in the Abukuma River for most species as of April 2021, reputational damage remains problematic for the carp aquaculture industry in particular. For wild species, one of the main issues in the contamination of salmonids eating preys that feed themselves on litter and fungi from contaminated forests, thereby demonstrating radiocesium accumulation along the food web (Wada et al., 2019).

## 9. Remobilisation of radionuclides from reservoir and pond sediment



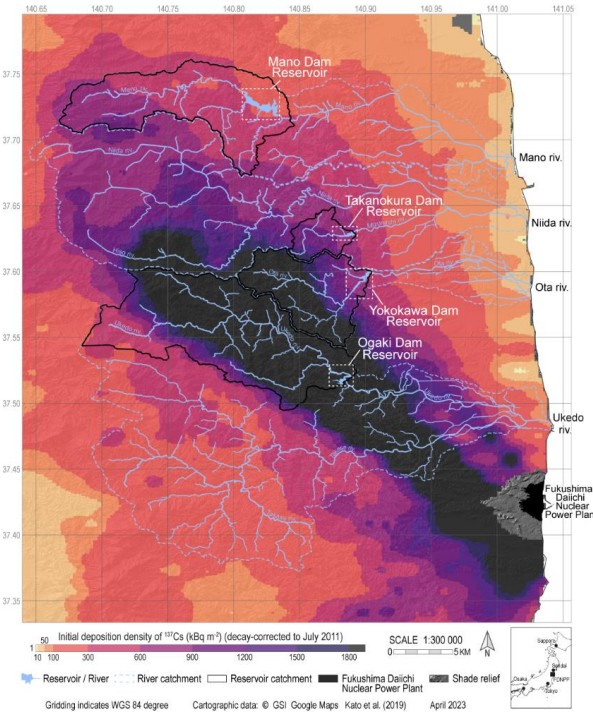

**Figure 5. Main dam reservoirs in the radioactive pollution plume of Fukushima Prefecture.**

Dam reservoirs were shown to act as a sink for radionuclides where they have been accumulating for more than one decade since the FDNPP accident (Sakai et al., 2021). In addition to the problems associated with the accumulation of contaminated sediment in these reservoirs, concerns were raised regarding the possible remobilisation of radionuclides from sediment to the water column under anaerobic conditions. Dissolved $^{137}$Cs concentrations found in sediment-pore water from two highly contaminated reservoirs (i.e., Ogaki Dam and Yokokawa Dam) of the Fukushima-impacted area (3–66 Bq L$^{-1}$) were one to two orders of magnitude higher than in reservoir water, showing evidence of the remobilisation of bioavailable $^{137}$Cs from sediment (Funaki et al., 2021). Furthermore, these authors identified a competitive ion exchange process between $^{137}$Cs and NH$_4^+$ via a highly selective interaction with the frayed edge sites of phyllosilicate minerals, leading to a very high variability of solid-liquid partition coefficient (K$_d$) values of sediment-pore water. The continuous supply of $^{137}$Cs -contaminated sediment from the upper catchment prevailed over the diffusive flux of $^{137}$Cs from sediment to overlying water.

Accordingly, reservoirs used for irrigation were shown to provide a perennial source of $^{137}$Cs both in particle-bound and dissolved forms, in response to resuspension and desorption processes. Furthermore, a control was identified between reservoir outflow water temperature and dissolved $^{137}$Cs activities in water. This further demonstrated that desorption of $^{137}$Cs from sediment is due to the exchange with cations such as NH$_4^+$ generated by biological activities. Furthermore, dissolved $^{137}$Cs concentrations in the outflow water exhibited seasonal variations with an increasing trend in summer (Kubota et al., 2022).





Funaki et al. (2020) investigated [137]Cs concentrations in input and output water from Ogaki Dam Reservoir (2014–2019), and they demonstrated that dissolved [137]Cs concentrations were significantly higher in outflow than in inflow water. They also calculated the mass balance of [137]Cs in the reservoir and showed that dissolved [137]Cs outputs were significantly higher than the inputs, and they estimated that 32%–40% of the dissolved [137]Cs in the output water was produced in the reservoir. It therefore represents a source of bioavailable dissolved [137]Cs,

with 0.04–0.09% of the [137]Cs accumulated in the reservoir sediment being eluted to the overlaying water each year. Similar results have been obtained from another reservoir (i.e. Matsugabou Dam) of the Fukushima Prefecture (Hayashi and Tsuji, 2020).

Furthermore, in the ponds of Okuma Town from 2015 to 2019 (Konoplev et al., 2021;Wakiyama et al., 2019), a decline in both particulate and dissolved [137]Cs activity concentrations was revealed. The decline rate constants

for the particulate [137]Cs activity concentration were found to be higher than for the dissolved [137]Cs activity concentration. In terms of seasonality, the dissolved [137]Cs concentrations were higher from June to October, depending on the specific pond and year, most likely due to temperature dependence of [137]Cs desorption from frayed edge sites of micaceous clay minerals. The apparent $K_d$([137]Cs) in the suspended sediment water system was observed to decrease over time. It was hypothesized that this trend was associated with the decomposition of

glassy hot particles.

This outlines questions regarding the interest of removing contaminated sediment accumulated in reservoirs in order to limit [137]Cs desorption and allow the safe resumption of agricultural water use. Another problem is related to the contamination of fish found in these ponds, as [137]Cs levels found in fish collected in 2015–2016 in four ponds in the DTRZ nearby FDNPP were found to be higher than in forest rivers of the zone and they

systematically exceeded by one to three orders of magnitude (up to 15,700 Bq kg[-1]) Japanese regulatory limit (100 Bq kg[-1]). This further demonstrates radiocesium bioaccumulation through the food web around bottom sediment in the ponds (Wada et al., 2019). This biomagnification process was observed in lakes and not in rivers (Ishii et al., 2020a). Even outside of the DTRZ and 5 years later, by 2020, despite remaining below the 100 Bq kg[-1] value, [137]Cs continued to exceed the detection limits of ca. 7 Bq kg[-1] in 92.7% of fish samples collected from

lakes and ponds of the Fukushima Prefecture where radiocesium has accumulated and may be progressively eluted from sediment (Wada et al., 2022). A peak in [137]Cs activities in lake fish was observed in Summer, which may reflect the preferential remobilisation of [137]Cs from sediment during this season due to the higher concentrations of $NH_4^+$ observed in bottom waters, although it may also be attributed to the higher feeding rates of fish observed during this part of the year (Matsuzaki et al., 2021).

**10. Conclusions**

Twelve years after the FDNPP accident, unprecedented decontamination works were completed across a wide area in Japan and their effectiveness could be demonstrated through continuous research and monitoring efforts implemented by numerous Japanese research groups and their foreign counterparts. Nevertheless, it remains important to continue environmental monitoring activities initiated after the accident using optimized spatio-

temporal approaches and novel indicators if necessary. Of note, data collection started earlier after the FDNPP accident compared to the situation in Chernobyl, and data was more comprehensive and shared in a more open way in Japan (Hashimoto et al., 2022c;Ishii et al., 2020b;Hashimoto et al., 2020b) although further improvements remain possible.



Based on this post-accidental experience, feedback can be provided to a wide range of communities, to improve
our preparedness in potentially affected regions (e.g., those located in the vicinity of nuclear power plants or
those that may be affected by contaminant deposition following other industrial accidents) in the future
(Hashimoto et al., 2022c). For instance, to be effective, the removal of the organic matter layer in deciduous
forests should be conducted rapidly after the accident, which is now too late in the case of Fukushima. As the
forest cover is too large to be fully decontaminated, in addition to the 20-m buffer zones along the forest edges,
priority could be given to the decontamination of the so-called *satoyama* zones (Hashimoto et al., 2022c). In all
cases, providing added value to the contaminated forest biomass is a real issue and it still requires the further
development and consolidation of various economic and technological approaches.

Runoff and river systems were shown to provide significant pathways of radiocesium redistribution.
Accordingly, the expected increased frequency of typhoons may be of concern, as these events may lead to
widespread flooding and significant forest disturbance (i.e., tree fall and associated landslides) (Morimoto et al.,
2021) and the associated $^{137}$Cs transfers. The potential impact of forest fires that may occur more frequently in
the Fukushima region in response to increasing fuel load and global change should also be investigated, as they
may lead to the release of $^{137}$Cs into the local atmosphere as investigated in the Chernobyl-affected region
(Evangeliou et al., 2014). Finally, the spatial pattern of deposition and the subsequent redistribution of
microparticles containing radiocesium that were found in different environmental compartments and that may
lead to specific health risks if they are inhaled should also be further investigated (Fueda et al., 2023).

### Data availability

All the data provided in this review article can be accessed directly in the referenced publications or URL.
Spatial layers displayed in Figures of the current manuscript are freely available on Zenodo (Evrard et al., 2023).

### Author contributions

Olivier Evrard took the initiative and the lead to write this review article. Thomas Chalaux Clergue drew the
maps and improved earlier map versions prepared by Olivier Evrard. Thomas Chalaux Clergue, Pierre-Alexis
Chaboche, Yoshifumi Wakiyama and Yves Thiry revised the manuscript and contributed to the text.

### Competing interests

Olivier Evrard is a member of the editorial board of SOIL.

### Acknowledgements

The support of CNRS (Centre National de la Recherche Scientifique, France), CEA (Commissariat à l'Energie
Atomique et aux Energies Alternatives, France) and the Institute of Environmental Radioactivity (IER) of the
Fukushima University in the framework of the International Research Project – IRP – MITATE Lab is gratefully
acknowledged.

### Financial support

This research benefitted from the support of the AMORAD (ANR-11-RSNR-0002) project (ANR, Agence
Nationale de la Recherche, Programme des Investissements d'Avenir). This work was also supported by ERAN
(Environmental Radioactivity Research Network Center) grants I-21-22 and I-22-24. Thomas Chalaux Clergue



obtained a PhD fellowship from CEA, France, and a JSPS grant to spend his second year of PhD at Kyoto
      Prefectural University (Oct. 2022 – Sept. 2023). This work was also supported by Grant-in-Aid for JSPS Fellows
      Grant number 22F22712 (Pierre-Alexis Chaboche) and PE22708 (Thomas Chalaux Clergue).

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
