# Peer review of "Research and management challenges following soil and landscape decontamination at the onset of the reopening of the Difficult-To-Return Zone, Fukushima (Japan)"

_EGUsphere, 2023_

## Author Response (AR1)

**Reply to editor's and reviewers' comments**

| Editor | |
|---|---|
| Both reviewers have found the manuscript of interest and suitable for publication. However, both suggest some clarifications and improvements to be made in the manuscript before publcation. Please, address the comments and suggestions of the reviewers in a revised version of the manuscript and submit it for further evaluation. | Many thanks for this general comment and for the opportunity to revise our manuscript. |
| **Reviewer 1** | |
| This is a really excellent holistic synthesis of the decontamination carried out in Fukushima areas. The synthesis is comprehensive, balanced, and insightful. This study will be a key literature that tells us about the decontamination measures taken in Fukushima and their impact/consequences. | Many thanks for this overall positive and encouraging comment! |
| General comment:

Scope Agreed?

I believe the content of this manuscript is within the scope of SOIL; however, I feel that the manuscript should probably be a bit more SOIL-centric context, by at least inserting some texts explaining why soil matters and plays a key role in the decontamination in the radioactive contamination problem, also in the abstract, purpose and conclusion, if possible, in the main text as well. No need to change the structure of the manuscript. | Reference to soils has been added to the abstract (LL.18-20; L.25; L.49), the introduction (LL-63-66) and the conclusions (L.648, L.672). |
| Specific comments:

"Japanese characters"

The use of Japanese characters for some keywords is very unique. I like it. Probably, to accurately translate (convey) the key terms. Right? | Indeed, also to provide the original terms in case the readers would like to have access to the original and official text/documents in Japanese. |
| Line 65-67 (Lyons et al., 2020): typo

Too many (Lyons et al., 2020)s. | Sorry about that. This has been corrected (L.69). |
| Line 181, 799, "Fukushima Reprun"

Probably "Reprun Fukushima" would be correct. | Corrected (L.189 and in references). |
| Line 163 Board of Audit of Japan （会計検査院） | Just in case the readers would like to have |

| | |
|---|---|
| I don't understand why you emphasize this by with Japanese characters. | access to the original and official text/documents in Japanese. |
| Line 187 Clean Centre (クリーンセンター)

I don't understand why you emphasize this by with Japanese characters. | Just in case the readers would like to have access to the original and official text/documents in Japanese. |
| Line 354: The Japanese characters here.

These two Japanese characters mean "debris". The correct one is shown in line 790. | Sorry about that. Corrected (L.372). |
| Line 440: "fuel break"

This term would be new for readers of SOIL. Add an explanation or rephrase it. | Explanation added in the text (LL.461-462). |
| Line 567 "salmon"

I am not a specialist of fish, but "salmon" sounds like a sea fish. Is this correct? | Actually, salmon is considered "anadromous" as it lives in both fresh and salt water. |
| References still need to be refined in terms of format. (137)Cs, for example. Please check them again. | Done, references were double-checked and corrected when needed. |
| Finally, again, I thank the authors for this excellent synthesis. | Thanks again! |
| **Reviewer 2** ||
| ## General comments
This review, which aims to inform the rest of the world about the unparalleled and proactive contamination measures taken by Japan, will be of great value as a preparation for future nuclear disasters. While many of the materials are written only in Japanese, I believe that the situation is adequately grasped and the necessary information is widely presented. | Many thanks for this overall positive and encouraging comment! |
| However, there are a few points that could be improved before publication. | Agreed. The statement was |

| | |
|---|---|
| Although the authors state that they do not deal with caesium migration (L129), the first half of Section 5 is mainly about caesium migration, which is contradictory.
Rather, as it relates to the theme of this review, it is necessary to consider the resumption of agriculture and forestry and their social and economic impact.
For example, two discussions by Kimura on the forestry economics impact of the accident would be helpful. (Unfortunately, it is in Japanese.)
https://www.jstage.jst.go.jp/article/jjfs/103/1/103_13/_pdf/-char/ja
https://www.jstage.jst.go.jp/article/jjfs/105/3/105_96/_pdf/-char/ja | modified accordingly.

Agreed. The manuscript focus statement has been modified (LL.131-134.

Reference to these two articles on forest economics was added (Table 1). |
| From the number of citations, it would be inferred that a significant part of the description of forest impacts and measures relies on information from a book of Hashimoto et al. (2022).
Presumably there are similarly reviews of other sections (e.g. in the agricultural section) that the authors have referred to.
Therefore, not only a review of caesium migration (Table 1) but also a review of the social impact of caesium contamination in each landscape should be listed. | References to these studies were added to Table 1. |
| ## specific comments
L65 (Lyons et al. 2020) -> ICRP citation is appropriate here. | Added (L.69) |
| Section on agricultural land (L228-272)
Measures, additional contamination and current situation of transfer factor in agricultural landscapes are well reviewed.
However, an explanation of the resulting actual concentrations in crops and the evolution of shipping restrictions would be essential. | This information has been added (LL.280-288). |
| L328 It should be mentioned that there are no restrictions on its use as a building material. | Added (L.345). |
| L348 Girdling is certainly worth trying as a means of reducing concentrations and increasing timber utilization.
However, it should be noted that it is not expected to be a solution, given the additional costs involved in Japan's difficult forestry management situation and the fact that pollution of cedar and other building materials is not a major problem.
As mentioned in the general comments, priority will be given to pointing out issues related to the forestry economy, etc. | This assertion has been nuanced to reflect better this debate (L.367). |

**Reply to editor's and reviewers' comments**

| | |
|---|---|
| L354 Confirmation on Japanese language errors. "瓦礫" -> "板葺". | Corrected (L.372). |
| Table 2 "Sawdust for mushroom cultivation" -> "Sawdust medium for mushroom cultivation" | Corrected (L.382) |
| L403 It should be mentioned that wood-logs have more stringent standards than sawdust and that the concentration of sawdust can be adjusted by changing the formulation. | Added (LL.422-424). |
| L411 Kobayashi -> Kanasashi et al 2020 https://www.sciencedirect.com/science/article/pii/S0265931X19307374 | Reference added (L.432). |
| L417 It is unclear which parts are distinguished as (i) and (ii) | This has now been clarified in the text (L.440). |
| L445 Hashimoto et al. state that the reason for the increase in wild boar and deer populations is not solely due to a decrease in the number of hunters (capture pressure). Fig. 6.14 in Hashimoto et al. shows that the number of captures has increased in line with the decrease in the number of hunters. Although the graph does not show the number of individuals, it should be understood that the estimated number of individuals has increased as well, and that the increase in the number of individuals has exceeded the pressure of captures. The 137th reference cited in Hashimoto et al. has been revised and the content changed, but previous versions can be downloaded from the following link. https://www.env.go.jp/nature/choju/plan/plan3-2a/chpt2.pdf (Japanese) | Many thanks, this has now been clarified in the text (LL.467-468). |